# *Dek504* Encodes a Mitochondrion-Targeted E+-Type Pentatricopeptide Repeat Protein Essential for RNA Editing and Seed Development in Maize

**DOI:** 10.3390/ijms23052513

**Published:** 2022-02-24

**Authors:** Zheyuan Wang, Weiwei Chen, Song Zhang, Jiawen Lu, Rongrong Chen, Junjie Fu, Riliang Gu, Guoying Wang, Jianhua Wang, Yu Cui

**Affiliations:** 1Beijing Innovation Center for Crop Seed Technology, Ministry of Agriculture and Rural Affairs, Key Laboratory of Crop Heterosis Utilization, Ministry of Education, College of Agronomyand Biotechnology, China Agricultural University, Beijing 100193, China; wangzheyuan@cau.edu.cn (Z.W.); 15501062618@163.com (W.C.); zhangsong0115@163.com (S.Z.); lujiawen@cau.edu.cn (J.L.); rilianggu@cau.edu.cn (R.G.); 2Institute of Crop Sciences, Chinese Academy of Agricultural Sciences, Beijing 100081, China; 670702crr@163.com (R.C.); fujunjie@caas.cn (J.F.); wangguoying@caas.cn (G.W.)

**Keywords:** E+-type, kernel development, maize, mitochondria, NAD3, PPR, RNA editing

## Abstract

In flowering plants, RNA editing is a post-transcriptional process that selectively deaminates cytidines (C) to uridines (U) in organellar transcripts. Pentatricopeptide repeat (PPR) proteins have been identified as site-specific recognition factors for RNA editing. Here, we report the map-based cloning and molecular characterization of the defective kernel mutant *dek504* in maize. Loss of *Dek504* function leads to delayed embryogenesis and endosperm development, which produce small and collapsed kernels. *Dek504* encodes an E+-type PPR protein targeted to the mitochondria, which is required for RNA editing of mitochondrial NADH dehydrogenase 3 at the *nad3*-317 and *nad3*-44 sites. Biochemical analysis of mitochondrial protein complexes revealed a significant reduction in the mitochondrial NADH dehydrogenase complex I activity, indicating that the alteration of the amino acid sequence at *nad3*-44 and *nad3*-317 through RNA editing is essential for NAD3 function. Moreover, the amino acids are highly conserved in monocots and eudicots, whereas the events of C-to-U editing are not conserved in flowering plants. Thus, our results indicate that *Dek504* is essential for RNA editing of *nad3*, which is critical for NAD3 function, mitochondrial complex I stability, and seed development in maize.

## 1. Introduction

RNA editing is a post-transcriptional modification occurring in vascular plant organelles through a deamination reaction at a specific nucleotide cytidine (C), which is converted to uridine (U) [1,2]. The major consequence of RNA editing is that the amino acid (aa) sequence differs from that encoded from the genome sequence. Typically, the amino acids encoded by edited mRNAs are more conserved than the genome-encoded ones [3]. Currently, RNA editing is viewed as a remedial DNA mutation at the RNA level to restore the evolutionarily conserved amino acids [3,4,5].

In angiosperms, most organelle transcripts harbor hundreds of editing sites. However, when plant mitochondrial RNA editing was discovered over 30 years ago, little was known regarding the molecular apparatus involved in identifying the correct C editing targets [6,7]. In 2005, Chlororespiratory Reduction 4 (CRR4), which possesses a pentatricopeptide repeat (PPR) motif, was found to be essential for the RNA editing of the chloroplast *ndhD* gene [8]. Since then, members of the PPR protein family have been identified as RNA editing factors required for site-specific C-to-U conversions in chloroplasts or mitochondria [9,10,11,12,13].

PPR proteins are present in all eukaryotes and constitute one of the largest protein families, with over 400 members in most species [14,15]. Based on the structure of their PPR motifs, these proteins are divided into two major subclasses: (i) P-PPR proteins, which only harbor canonical P motifs of 35 amino acids, and (ii) PLS-PPR proteins, which harbor the canonical P motif, a long (L, 35–36 aa) motif, and a short (S, 31–32 aa) motif, forming tandemly repeated P-L-S triplets [16,17]. The majority of the PLS-PPR proteins contain additional C-terminal E, E+, and DYW domains [16]. P-PPR proteins are mainly involved in various aspects of organelle RNA processing, whereas PLS-PPR proteins are almost exclusively related to C-to-U conversion [18].

In addition to PPR proteins, C-to-U RNA editing involves proteins from diverse families, including multiple organelle RNA editing factors/RNA-editing factor interacting proteins (MORFs/RIPs) [19,20], organelle RNA recognition motif-containing proteins (ORRMs) [21], protoporphyrinogen IX oxidase 1 (PPO1) [22], and organelle zinc finger proteins (OZ1) [23]. Accumulating evidence indicates that PPR proteins regulate C-to-U RNA editing in plant organelles via forming an editing complex with other editing factors [9,19,24,25,26,27,28,29].

The maize NB mitochondrial genome harbors 493 editing sites in 58 mitochondrial genes [28], with 399 sites in 22 genes encoding the subunits of electron transport chain (ETC) complexes (I–V) and cytochrome c biogenesis proteins (e.g., *nad1*, *nad2*, *nad3*, *nad4*, *nad4L*, *nad5*, *nad6*, *nad7*, *nad9*, *cob*, *cox1*, *cox2*, *cox3*, *atp1*, *atp4*, *atp6*, *atp8*, *ccmB*, *ccmC*, *ccmFC*, and *ccmFN*) [30]. To date, 19 PLS-PPR proteins have been shown to be essential for the C-to-U RNA editing of mitochondrial transcripts in maize (Appendix A). Of these, 5 proteins, namely PPR27, DEK53, DEK55, EMP5, and EMP21, are involved in editing at multiple sites, and the remaining 14 are involved in the editing at one or a few specific sites [9,26,27,28,29]. The loss-of-function of these PPR proteins abolishes editing or reduces editing efficiency, ultimately affecting the assembly and activity of the related mitochondrial complexes. Simultaneously, the alternative oxidase (AOX) pathway is activated, indicating defective electron transport in the cytochrome c pathway [31,32]. The mitochondria serve as the energy supply and redox regulation center in cells [33], which convert the chemical energy stored in carbon substrates to adenosine triphosphate (ATP) via ETC [33,34,35]. Typically, the impairment of mitochondrial function delays embryonic and endosperm development and arrests basal endosperm transfer layer (BETL) development, ultimately producing mutant phenotypes with empty pericarps (emp) and defective or small kernels (dek or smk, respectively) [9,29,36,37].

PPR proteins play pivotal roles in RNA editing to maintain mitochondrial function during embryonic and endosperm development. In the present study, a new maize *dek* mutant, *dek504*, exhibiting a small and delayed kernel development phenotype, was characterized. *DEK504* encodes a mitochondrion-targeted E+-type PPR protein that is involved in C-to-U editing at specific positions of NADH dehydrogenase gene 3 (*nad3*-44 and *nad3*-317). Defects at these editing sites impede the assembly and stability of mitochondrial complex I, thus impairing the structure and function of mitochondria and arresting the development of embryos and endosperm in maize.

## 2. Results

### 2.1. Genetic and Phenotypic Characterization of the dek504-ref Mutant

A spontaneous mutant exhibiting a defective kernel phenotype was isolated from an inbred line *14d3147* and named *dek504-ref*. The segregation ratio of normal to defective kernels on the self-crossed ears of *dek504-ref* heterozygote was consistent with the Mendelian separation ratio of 3:1 (χ^2^ = 0.530 < χ^2^_0.05_ = 3.84; Figure 1a), indicating that *dek504-ref* is a monogenic recessive mutation. This result was confirmed in the F_2_ populations generated from the crosses of *dek504-ref/*+ with B73, Mo17, and Zheng58 inbred lines (Appendix A).

Contrary to the wildtype (WT) maize, *dek504* developed small kernels and appeared collapsed at maturity, with dramatic size and weight reduction (Figure 1b). The 100-kernel weight of *dek504* was 45.7% of the weight of WT (Figure 1g). In longitudinal and transverse sections, *dek504* kernels showed distinctly wrinkled embryo and endosperm structures (Figure 1c,d). Approximately 65% of the *dek504* kernels germinated, although the germinated seedlings showed retarded growth (Figure 1f). While the WT seedlings formed a primary root and several seminal roots after 7 days of germination, *dek504* seedlings only formed a primary root. The length of the primary roots and shoots was significantly shorter in *dek504* seedlings than in WT seedlings (Figure 1e). Moreover, some of the germinated *dek504* seedlings completed the entire growth period and produced small seeds with a visible collapsed seed coat (Appendix A).

Scanning electron microscopy (SEM) revealed that the starch granules (SGs) in WT internal endosperm cells were smooth and tightly packed. However, SGs in *dek504* endosperm cells were loosely and irregularly packed and exhibited many holes on the surface (Figure 1h). In mature kernels, the total starch and protein contents in *dek504* endosperm were, respectively, 8.7% and 27.5% lower than those in WT endosperm (Figure 1i,j). Overall, the *dek504* mutation delayed plant growth and seed development as well as lowered seed nutrient reservoir accumulation.

The developing kernels of *dek504* and WT were observed at 12, 15, and 18 days after pollination (DAP). The *dek504* kernels could be distinguished from the WT kernels as early as 12 DAP, characterized by their small size and translucent appearance with white top (Figure 2a). Paraffin-embedded sections showed that the WT embryos developed visible scutellum, coleoptile, leaf primordium (LP), and shoot apical meristem (SAM) and reached the L2 (second leaf primordia) stage at 12 DAP (Figure 2b); whereas *dek504* embryos arrested and remained at the transition stage, without visible differentiation in LP and SAM. The WT embryo developed 3–4 leaf primordia and complete structures with clearly visible SAM and root apical meristem (RAM) at 15 and 18 DAP, reaching late embryogenesis stage (Figure 2c–e). By contrast, the *dek2849* embryos reached the L1 stage with one leaf primordium and SAM but without RAM at 15 DAP. At 18 DAP, the mutant embryos remained at the L1 stage and showed no significant sign of RAM development (Figure 2c–e). In addition, compared with WT kernels, which had starch-filled endosperms, the *dek504* kernels had small and underdeveloped endosperms, with a gap observed between the endosperm and seed coat at 12 DAP (Figure 2b). The *dek504* endosperm development was slower than that of the WT at 15 and 18 DAP (Figure 2c,d). The WT kernels showed extensive cell wall ingrowth in the basal endosperm transfer layer (BETL) cells at 15 DAP, whereas *dek504* kernels comprised small cells and showed little cell wall ingrowth in BETL (Figure 2f). Overall, the *dek504-ref* mutation affected both embryonic and endosperm development.

### 2.2. Positional Cloning of dek504

The *dek504-ref* heterozygotes were crossed with the B73 inbred line to generate an F_2_ population for the map-based cloning of *dek504*. Homozygous mutant and WT kernels from segregated F2 ears at 15 DAP were used for bulked segregant RNA-Seq (BSR-Seq), which initially mapped the *dek504* gene to a 6.7 Mb interval on chromosome 7 (Appendix A). After characterizing 6192 homozygous mutant kernels using seven markers within this 6.7 Mb interval (Appendix A), the mapping region was narrowed to a 60.3 Kb region containing four putative protein-coding genes, namely G1-*Zm00001d022392*, G2-*Zm00001d022393*, G3-*Zm00001d022394*, and G4-*Zm00001d022395*, according to the B73 reference genome (AGPv4) (Figure 3a). Genomic sequencing revealed the same sequence of G1 and G2 open reading fragments (ORFs) between WT and *dek504* but revealed a 6.9 Kb deletion in the *dek504* genome from 176,672,153 to 176,679,118 bp. This deletion led to the complete loss of the *G4* ORF and partial loss of the *G3* ORF (Figure 3a–c). Reverse transcription–polymerase chain reaction (RT-PCR) failed to amplify the transcripts of *G3* and *G4* ORFs, further confirming the presence of this deletion.

To determine the causal gene for *dek504*, the G3 (*g3-mut*) and G4 (*g4-ems*) mutants were characterized. *g4-ems* was an EMS mutant, containing a single-nucleotide substitution (C to T) at 396 bp of the G4 ORF, resulting in a premature stop codon and a truncated protein (Appendix A). *g3-mut* was a spontaneous mutant, carrying a 3017 bp Gypsy31-ZM_LTR insertion at 134 bp of the G3 ORF (Figure 3b,d). The heterozygous *g4-ems* and *g3-mut* mutants were crossed with the heterozygous *dek504-ref* mutant for allelic testing, which revealed that the kernels generated from the cross between *dek504-ref* and *g3-mut* exhibited a 3:1 segregation ratio of the WT and *dek* phenotypes (681:211, χ^2^ = 2.509 < χ^2^_0.05_ = 3.84; Figure 3f); however, all kernels generated from the cross between *dek504-ref* and *g4-ems* exhibited the WT phenotype (Appendix A). Therefore, G3 (*Zm00001d022394*) but not G4 is the causative gene of *dek504*.

### 2.3. Dek504 Encodes a Mitochondrion-Targeted E+-Type PPR Protein

RACE experiments revealed that the *Dek504* ORF in WT was 1830 bp. However, in the *dek504* mutant, the genome deletion led to the loss of ORF sequence from +465 to +1830 bp, and the addition of a 157 bp transcript from the flanking genome sequence resulted in a mutant ORF of 621 bp (Appendix A). *DEK504* was predicted to encode a 66 kD PPR protein. Motif prediction analysis using PLANTPPR (https://ppr.plantenergy.uwa.edu.au/, accessed on 27 April 2019) and TPRpred (http://tprpred.tuebingen.mpg.de/tprpred, accessed on 27 April 2019) [17,38] revealed that DEK504 contained 12 PPR motifs, one E1/E2 domain, and one E+ domain; therefore, the protein was classified as an E+-type PLS-PPR protein (Figure 4a). The truncated DEK504 protein in the *dek504* mutant lacked the last 10 PPR motifs, the E1/E2 domain, and the E+ domain (Figure 4a).

Phylogenetic analysis based on the protein sequence of 12 DEK504 homologs from 12 species revealed that these proteins diverged into two separate clades of monocots and eudicots (Figure 4b). DEK504 showed a higher similarity to monocot homologs (75.54~91.3%) than to eudicots (42.49~48.82%), with the highest similarity (91.3%) with the *Sorghum bicolor* homolog (Figure 4b). The detailed sequence alignment revealed highly conserved P/L/S/E domains among monocot DEK504 homologs (Appendix A).

TargetP (http://www.cbs.dtu.dk/services/TargetP/, accessed on 4 May 2019) and Predotar (http://urgi.versailles.inra.fr/predotar/predotar.html, accessed on 4 May 2019) predicted that DEK504 was localized in the mitochondria. Next, the *Dek504* ORF was fused to the gene encoding an enhanced yellow fluorescent protein (eYFP) driven by the CaMV 35S promoter and transformed into *Arabidopsis*. The fluorescent signal of DEK504-eYFP overlapped with the red fluorescence of the mitochondrial localization marker MitoTracker pBIN20-MT-YB [39], indicating that DEK504 was targeted to the mitochondria (Figure 4c).

Quantitative RT-PCR (qRT-PCR) revealed that *Dek504* was constitutively expressed in all tested organs, including roots, stems, leaves, bracts, tassels, silk, and spike stalk, with relatively higher expression in kernels (Figure 4d). During kernel development, *Dek504* expression was detected at 5 DAP; its expression gradually increased thereafter, peaking at 10 DAP, and then gradually decreased at the later stages (Figure 4d).

### 2.4. DEK504 Is Required for the C-to-U Editing of Mitochondrial nad3 Transcript

A previous study showed that PPR-E+ proteins are involved in the C-to-U editing of mitochondrial RNA [18]. Therefore, the editing sites of DEK504 were explored using the RNA-Seq of *dek504* mutant and WT kernels of 15 DAP. The mutation did not significantly affect the editing sites of mitochondrial transcripts. However, editing at *nad3*-44 (from 80% to 40%) and *nad3*-317 (from 94% to <1%) was dramatically decreased in *dek504* kernels. The DNA nucleotides of *nad3*-44 and *nad3*-317 sites were the same between *dek504* and WT kernels, excluding the possibility that polymorphisms (SNPs) affect editing efficiency. The RNA editing sites in WT and *dek504* were confirmed by RT-PCR, which demonstrated a reduction in editing from 70% to 8% at *nad3*-44 and from 75% to 0% on *nad3*-317 in the *dek504* kernel (Figure 5a). According to the codes defined by the combinatorial residues at positions 6 and 1 [15,40], the identity of the combinatorial codes of PPR repeats well matched the recognition sites around the RNA editing sites *nad3*-44 and *nad3*-317 (Figure 5b). Therefore, the DEK504 protein is necessary for RNA editing at *nad3*-44 and *nad3*-317 in maize.

Deficient editing at *nad3*-44 and *nad3*-317 in the *dek504* mutant led to a Leu-to-Pro conversion at NAD3-15 and a Phe-to-Ser change at NAD3-106 (Figure 5a). gDNA sequence alignment showed that both editing sites encoded “C” in monocots, with the exception of “T” at *nad3*-317 in rice; however, both editing sites encoded “T” in lower plants, and “C” or “T” remained in eudicots. cDNA sequence alignment of NAD3 orthologs revealed that these two amino acids were conserved in bryophytes, dicots, and monocots (Figure 5c), implying their significance for the functional integrity of NAD3 proteins.

### 2.5. Dek504 Mutation Affects Mitochondrial Complex I Assembly and NADH Dehydrogenase Activity

Previous studies have reported that editing defects in mitochondrial transcripts affect the activity of mitochondrial respiratory complexes and lead to mitochondrial dysfunction [41,42,43]. To further investigate whether changes in *nad3* editing efficiency affect mitochondrial complex I formation and function, mitochondrial proteins were isolated from *dek504-ref* and WT kernels and analyzed by blue native polyacrylamide gel electrophoresis (BN-PAGE). There was a slight decrease in mitochondrial complex I and super complex I+III_2_ formation, but a slight increase in complex III and V formation in *dek504-ref* compared with that in WT (Figure 6a). The NADH dehydrogenase activity of complex I was quantitatively evaluated using ImageJ (https://imagej.nih.gov/ij/, 1.6.0_24, accessed on 4 October 2019) [44], which revealed a 43% decrease in *dek504-ref* (Figure 6b). Therefore, the reduction in editing efficiency at *nad3* impeded complex I assembly and activity in *dek504*.

To assess the impact of defective RNA editing at *nad3* on the accumulation of respiratory chain complexes, the concentration of mitochondrial complex-related proteins, including NADH dehydrogenase subunit 9 (Nad9), CytC (cytochrome c), cytochrome oxidase subunit 2 (Cox2), and α subunit of ATPase (αATPase), was detected via immunoblotting. The concentration of complex I subunit Nad9 in *dek504* was decreased to 50% of that in WT, whereas the concentration of proteins forming complexes III and V was increased (32%, 63%, and 70% increase for Cytc, Cox2, and ATPase-B, respectively) in *dek504* (Figure 6c). Therefore, in *dek504*, the structural components and activity of complex I were downregulated, but those of the other complexes were upregulated, due perhaps to a compensatory metabolic pathway.

### 2.6. Mitochondria Morphology Was Affected and an Alternative Respiratory Pathway Was Activated in the dek504-ref Mutant

Abnormalities in any mitochondrial process can lead to mitochondrial dysfunction, such as mtROS generation and apoptosis [45]. Transmission electron microscopy (TEM) indicated that in WT, the mitochondria formed a continuous closed surface with complex morphology and cristae with densely folded corrugated inner membrane. In contrast, the internal structure of mitochondria in *dek504-ref* was unclear, lacking the typical cristae, and the mitochondrial matrix was lighter (Figure 7a–d). These results indicate that *Dek504* is necessary for the establishment of a normal mitochondrial structure.

Alternative oxidase (AOX) can be induced as an alternative electron transport pathway when the primary respiratory cytochrome c pathway is impaired [46,47]. qRT-PCR was used to evaluate the expression of *AOX* genes *AOX1* (*Zm00001d017727*), *AOX2* (*Zm00001d002436*), and *AOX3* (*Zm00001d002434*) in *dek504*. The transcript levels of *AOX2* were markedly increased in *dek504-ref* compared with those in WT (Figure 7e). Consistently, AOX2 protein expression was dramatically increased in *dek504-ref* (Figure 6c). Therefore, the loss-of-function of DEK504 suppressed complex I activity, subsequently impairing the mitochondrial respiratory pathway and inducing an alternative respiratory pathway in *dek504-ref*.

## 3. Discussion

### 3.1. DEK504 Encodes an E+-Type PPR Protein Involved in Editing at nad3-44 and nad3-317

Previous studies have shown that PPR proteins serve vital roles in post-transcriptional processes, including RNA editing, splicing, cleavage, and translation in the mitochondria and chloroplasts [14]. There are 49 E+-type PPR genes in maize, of which four genes, namely *Dek36*, *Emp9*, *Dek40*, and *Smk6*, have been previously characterized [37]. All these genes are involved in kernel development, and the loss function of these genes resulted in different phenotypes, with *empty pericarp* for *Emp9*, *defective kernel* for *Dek36* and *Dek40*, and *small kernel* for *Smk6.* In the present study, we characterized another E+-type PPR protein, DEK504, which is involved in RNA editing in maize mitochondria. The loss of function of DEK504 resulted in the *dek* phenotype of the kernel. Although all five known E+-type PPR proteins exhibit C-to-U RNA editing of mitochondrial transcripts, they differ in terms of the editing genes and sites: *Dek36* at *atp4*-59, *ccmFN*-302, and *nad7*-383; *Dek40* at *nad5-1916*, *nad2-26*, and *cox3-314*; *E**mp9 at ccmB-43 and rps4-335*; *Smk6* at *nad1-740*, *nad4L-110*, *nad7-739*, *mttB-138*, and *mttB-139*; and *Dek504 at nad3*-44 and *nad3*-317 [37,41,48,49]. Therefore, the present study unveiled a novel mitochondrion-localized E+-type PPR protein (Figure 4c) involved in C-to-U RNA editing at two novel sites, namely *nad3*-44 and *nad3*-317 (Figure 5a).

Previous studies have shown that some E- and DYW-subgroup PPR proteins can interact with the MORF/RIP family proteins to realize their editing functions [9,26,27,28,29]. However, the interaction between the E+-PPR protein and other editing factors in maize has not been suggested. Here, we applied a yeast two-hybrid (Y2H) assay to detect the interactions of DEK504 with eight MORF and four organelle RNA recognition motif (ORRM)-containing proteins; however, DEK504 did not interact with any ZmMORF and ZmORRM protein (Appendix A).

### 3.2. Dek504 Editing Sites Are Important for Mitochondrial Function

Mitochondrial RNA editing has been documented in all major groups of land plants, except bryophytes. In the present study, *Dek504* was shown to be essential for C-to-U editing at *nad3*-44 and *nad3*-317. gDNA and cDNA sequence analysis revealed that both editing sites were conserved in monocots, with the exception of *nad3*-317 in rice (Figure 5c). Conserved editing sites among monocot species, including *nad7*-836 site by SMK1; *nad3*-61, *nad3*-62, and *cox2*-550 by DEK10; *nad3*-247 and *nad3*-275 by DEK39; *nad7*-77, *atp1*-1292, and *atp8*-437 by EMP21; and *ccmF_C_*-799 and *nad2*-677 by EMP17, have already been revealed previously [12,28,36,50,51]. However, RNA editing at *nad3*-44 and *nad3*-317 is not conserved in dicots and can be divided into different types (Figure 5c). Although the conservation of *nad3*-44 and *nad3*-317 editing sites differs between monocots and dicots, Leu and Phe were encoded at NAD3-15 and NAD3-106, respectively, and conserved across bryophytes, dicots, and monocots, implying that these residues are important for the functional integrity of the NAD3 protein.

In plants, respiratory metabolism occurs via mitochondrial ETC (mETC), which involves four multisubunit complexes (I–IV) and cytochrome c [34]. Nad3 encodes subunit III of NADH dehydrogenase as part of complex I, which is the primary entry point for electrons into the mETC [52]. The lack of editing at *nad3*-44 and *nad3*-317 as a result of the *dek504* mutation resulted in the surrogate mutations of Leu to Pro-15 and Phe to Ser-106, respectively. Transmembrane helix prediction indicated that ZmNAD3 possesses three transmembrane helices, with Leu-15 and Phe-106 being located in TMH1 and TMH3, respectively (Appendix A).

Mitochondrial Nad3 transcripts of maize contain 66 C-to-U editing sites [53], and only two E-subclass PPR proteins and one DYW-PPR protein have been identified to be involved in RNA editing at these four sites. The *Dek10* mutation impeded C-to-U editing at *nad3*-61 and *nad3*-62, resulting in an amino acid substitution from Leu to Pro-21, which is conserved among monocots and located in TMH1 [37]. Loss of *Dek39* function affected C-to-U editing at *nad3*-247 and *nad3*-275, resulting in a conversion from Ser to Pro-83 and Phe to Ser-92, which are located in TMH2 and TMH3, respectively [51]. EMP21 is required for editing at multiple sites, including *nad3*-275, which is also edited by *Dek39* [28,51]. These results indicate that the above five surrogate mutations may negatively impact the structural stability of NAD3 TMH, thus affecting NAD3 function. Taken together, these results indicate that C-to-U editing at *nad3*-44 and *nad3*-317 sites is important for NAD3 function.

Abnormal C-to-U editing in Nad3 transcripts affected the assembly and activity of mitochondrial complex I. Disrupted functions of complex I due to the lack of RNA editing at *nad3*-250 (*slg1*) [54], *nad3*-155, *nad3*-172, *nad3*-173, *nad3*-190, *nad3*-191 (*pps1*) [55], *nad3*-61, *nad3*-62 (*dek10*) [36], *nad3*-247, and *nad3*-275 (*dek39*) [51] have been detected. Our results showed that the NADH dehydrogenase activity of complex I in *dek504* was reduced to 57% of that in WT due to the disruption of editing at *nad3*-44 and *nad3*-317, which impaired mitochondrial electron transport and structural integrity (Figure 7a–d). The status of mitochondrial electron transport is a vital factor determining plant growth [56,57], and an alternative respiration pathway is activated to compensate for ATP deficiency when the canonical mitochondrial ETC is impaired [58]. In the present study, the transcript levels of *AOX* genes, particularly *AOX2*, were noticeably increased in *dek504*, indicating the activation of the alternative respiratory pathway. Thus, energy shortage may be the major cause of growth defects in *dek504*.

## 4. Materials and Methods

### 4.1. Plant Material and Growth Conditions

Mutants *dek504-ref* and *g3-mut* were derived from the inbred lines 14d3147 and 14d3010, respectively. The original name of the allele mutant *g4-ems* was EMS4-0a0e7, which was purchased from a maize EMS-induced mutant database (http://www.elabcaas.cn/memd/, accessed on 14 September 2019) [59]. The F2 population of 6192 mutant kernels was used to map the *dek504* locus in the B73 background. The maize plants were grown at Shunyi, experimental station of Institute of Crop Sciences, Beijing, China (116.60°E, 40.22°N) and Nanbin, the experimental station of the Institute of Crop Sciences, Sanya, Hainan Province, China (109.18° E, 18.36° N) under natural conditions. *Arabidopsis* (Columbia-0, Col-0) was grown at 22 °C under a 16 h/8 h light/dark cycle.

### 4.2. Light Microscopy of Cytological Sections, TEM, and SEM

The *dek504-ref* and WT kernels from a segregating F2 ear at 12, 15, and 18 DAP were prepared for cytological analysis according to previously published methods [60]. Paraffin-embedded sections were stained with toluidine blue and observed under the Nikon Ti microscope (Nikon, Tokyo, Japan). Immature WT and *dek504-ref* kernels at 15 DAP were treated for TEM. Their dry kernels were sectioned along the embryo, and SEM was used to observe the protein bodies, as described previously [61].

### 4.3. Protein and Starch Content Measurement

The endosperm was separated from the embryo and pericarp, dried to a constant weight, and pulverized with a cryogenic grinder. Protein content was measured according to a previously described protocol [62]. Starch was extracted and measured in 500 mg of powdered sample using an amyloglucosidase/a-amylase starch assay kit (Megazyme, Bray, Ireland), as described previously [63]. Three biological replicates were set for subsequent analyses.

### 4.4. Map-Based dek504 Cloning

Thirty mutant and normal individuals from the same ear of the F_2_ population were collected at 15 DAP and pooled for BSR-Seq analysis according to a published method [64]. An F_2_ population of 6192 mutant kernels was used for fine mapping with the molecular markers listed in Appendix A. Following gene annotation and resequencing, the corresponding DNA fragments of the candidate genes in the identified region in *dek504-ref* and WT-*ref* were amplified using KOD PLUS DNA polymerase (Toyobo, Osaka, Japan) and sequenced.

### 4.5. 3′-RACE

The full-length 3′-termini of the WT and *dek504-ref* transcripts were isolated using the SMARTerTM RACE cDNA Amplification Kit (Clontech, Mountain View, CA, USA) in accordance with the instructions of the manufacturer using the primers listed in Appendix A. The amplicons were cloned into the pEASY^®^-Blunt Simple Cloning Vector (TransGen, Beijing, China) and sequenced.

### 4.6. Subcellular Localization

The ORF of *Dek504* without the stop codon was amplified from the maize inbred line B73 and cloned into the expression vector pEarleyGate101 to construct the p35S::DEK504-eYFP plasmid using the gateway technology (Invitrogen, Carlsbad, CA, USA). This vector was transformed into *Arabidopsis thaliana* Col-0 using the floral dip method [65], and the lateral root hairs of the transgenic plants were used for YFP observation. MitoTracker Red (Invitrogen) was used to label the mitochondria. Fluorescent signals were observed under a confocal laser scanning microscope (Zeiss LSM700; Carl Zeiss, Oberkochen, Germany) using eYFP (514 nm excitation and 519–559 nm emission wavelength), mCherry (587 nm excitation and 590–630 nm emission wavelength), and Mito Tracker Red (581 nm excitation and 644 nm emission wavelength).

### 4.7. RNA Extraction, RT-PCR, and qRT-PCR

RNA was extracted using a commercial kit (Tiangen, Beijing, China), and residual gDNA was removed with RNAse-free DNAse I (NEB) treatment. Then, RNA was reverse transcribed into cDNA using the TransScript II One-Step gDNA Removal Kit and cDNA Synthesis SuperMix Kit (TransGen, Beijing, China) with Oligo (dT) primers. The primers used for RT–PCR and qRT-PCR are listed in Appendix A. Glyceraldehyde-3-phosphate dehydrogenase (GAPDH) was used as the internal control.

### 4.8. Mitochondrial RNA Editing Analysis

For RNA editing analysis, 30 seeds of *dek504* and WT (following pericarp removal) at 15 DAP were pooled for RNA-Seq, with three biological replicates. Total RNA from each sample was extracted using RNAprep Pure Plant Kit (DP432, Tiangen, Beijing, China) following the manufacturer’s protocol. The RNA integrity number and concentration were determined using the Agilent 2100 Bioanalyzer (Agilent, Santa Clara, CA, USA). RNA-Seq libraries were prepared using the Illumina Standard mRNA-Seq Library Preparation Kit (Illumina, San Diego, CA, USA) and sequenced to generate 150 bp paired-end reads on the Illumina HiSeq4000 (Illumina) platform. Adaptor sequences, unknown nucleotides > 5%, or percentage of reads with sequencing error rates < 1% were removed using a Perl script. Each sample of the clean reads with two biological replicates was first merged and aligned to the B73 reference genome (AGPv4) using HISAT2 (v2.1.0) [66]. Single-nucleotide polymorphic sites were identified by Samtools (v1.3.1) [67]. Editing frequencies of the same sites were calculated using Samtools mpileup (https://www.htslib.org, accessed on 4 May 2019). Sites covering over 10 reads were used to analyze the editing efficiency. Sites located within the gene-coding region were selected to analyze the changes in editing efficiency.

To analyze the editing efficiency at *nad3*-44 and *nad3*-317, mitochondrial cDNA was amplified, and the PCR products were cloned into the *pEASY* ^®^-Blunt Simple Cloning Vector (TransGen, Beijing, China). Ninety clones each from WT and *dek504-ref* PCR products were separately sequenced. The primers used are listed in Appendix A.

To exclude possible polymorphisms (SNPs) at the *nad3*-44 and *nad3*-317 sites in the *dek504* mutant and WT mitochondrial genomes, DNA of 10 kernels were extracted from *dek504*, WT-ref, and B73 each. A 357 bp DNA fragment of the *nad3* region was amplified using specific primers (nad3-F and nad3-R). DNA fragments were sequenced separately for each sample by ABI 3730XL sequencer.

### 4.9. Isolation and Analysis of Mitochondria Complexes and Western Blotting

Crude mitochondria were extracted from 15 DAP kernels of *dek504-ref* and WT (without pericarp) according to a previously described method [64]. For BN-PAGE and NADH dehydrogenase activity of complex I analysis, mitochondrial proteins (100 μg) were solubilized using the NativePAGE™ Sample Prep Kit (Invitrogen, Carlsbad, CA), as described previously [64].

Mitochondrial proteins (20 μg) extracted from WT and *dek504-ref* kernels at 15 DAP were used for Western blotting. Proteins were separated by SDS–PAGE, transferred to a polyvinylidene difluoride (PVDF) membrane, and incubated overnight at 4 °C with various target protein antibodies. Following incubation with the HRP-conjugated secondary antibody, the signals were visualized using the SuperSignalTM West Pico PLUS Chemiluminescent Substrate Kit (Thermo, Waltham, MA, USA) in accordance with the instructions of the manufacturer. The antibody dilutions used were 1:1000 for Cox2 (Agrisera), 1:1000 for NAD9 (PHYTOAB), 1:2000 for Cyt-c (Agrisera), 1:2000 for AOX (PHYTOAB), and 1:1000 for ATPase-B (Agrisera).

## Figures and Tables

**Figure 1 ijms-23-02513-f001:**
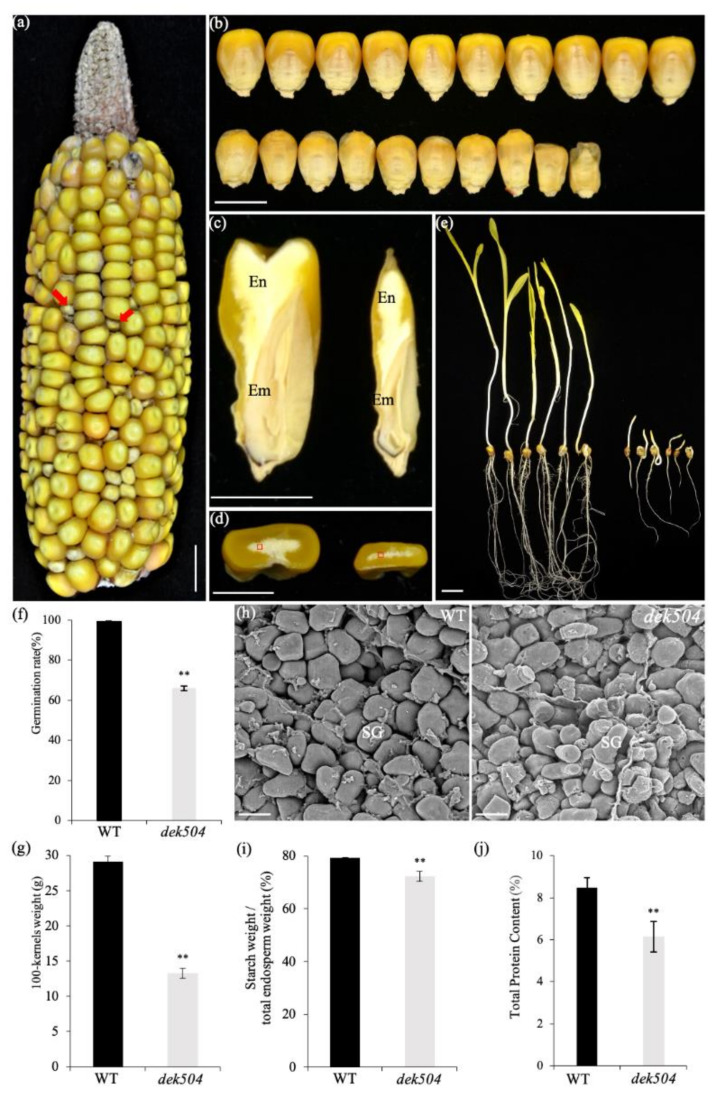
Phenotypic characteristics of the maize *dek504-ref* mutant. (**a**) Mature self-pollinated ear of *dek504-ref* heterozygous plant; red arrow indicates the *dek504* kernel. Bar = 1 cm. (**b**) Front view of wildtype (WT, up) and *dek504-ref* (down) kernels. Bar = 1 cm. (**c**,**d**) Longitudinal (**c**) and transverse (**d**) sections of WT (left) and *dek504-ref* (right) kernels. Bar = 0.5 cm. (**e**,**f**) Germination test (**e**) and rate (**f**) of WT and *dek504-ref* seedlings at 7 days after germination (DAG). Bar = 2 cm. (**g**) Comparison of the 100-kernel weight of randomly selected mature WT and *dek504-ref* plants. (**h**) Scanning electron micrographs of WT and *dek504-ref* mature endosperm. SG, starch granule. Bar = 10 µm (**i**,**j**) Starch (**i**) and total protein (**j**) content of WT and *dek504-ref* endosperm at the mature stage. Values indicate means ± SD (*n* = 3; ** *p* < 0.01, Student’s *t*-test).

**Figure 2 ijms-23-02513-f002:**
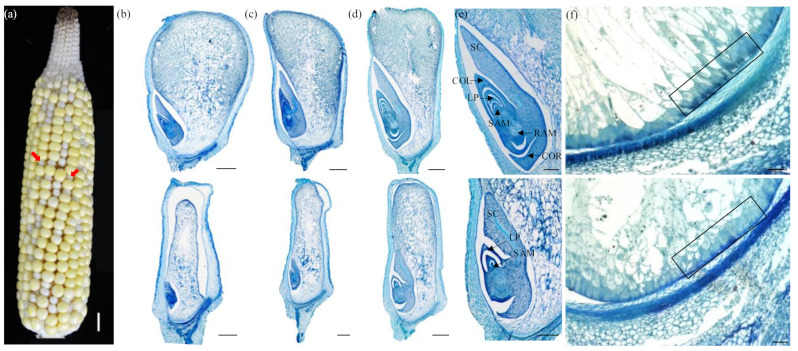
Comparison of developing kernels between wildtype (WT) and *dek504-ref* maize. (**a**) Self-pollinated *dek504-ref* heterozygote ears at 12 days after pollination (DAP). Bar = 1 cm. (**b**–**d**) Histological analysis of WT (above) and *dek504* (bottom) kernel at 12 (**b**), 15 (**c**), and 18 DAP (**d**). Bar = 1 cm. (**e**) Magnified image of Figure 2d. SC, scutellum; COL, coleoptile; LP, leaf primordium; SAM, shoot apical meristem; RAM, root apical meristem; COR, coleorhiza. Bar = 1 mm. (**f**) Comparison of the basal endosperm transfer layer (BETL) between WT and *dek504* at 15 DAP. Bar = 50 µm.

**Figure 3 ijms-23-02513-f003:**
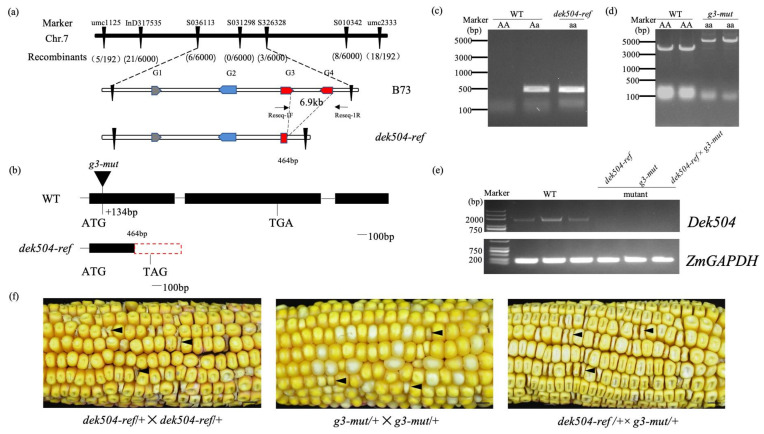
Cloning and identification of *dek504*. (**a**) Fine mapping of the *dek504* locus. The *dek504* locus was mapped to a 70 kb region on chromosome 7, harboring four putative genes. (**b**) Gene structure and mutation site of *Dek504*. (**c**,**d**) Genotype analysis of wildtype (WT), *dek504-ref* (**c**), and *g3-mut* (**d**) kernels. (**e**) Full-length open reading frame (ORF) amplification of *Dek504* expression in WT and *dek504* mutant alleles using RT-PCR. Expression was normalized against *ZmGAPDH*. (**f**) Heterozygous *dek504-ref* and *g3-mut* were used in the allelism test of *Dek504*. Black arrow indicates the mutant kernel. Bar = 1 cm.

**Figure 4 ijms-23-02513-f004:**
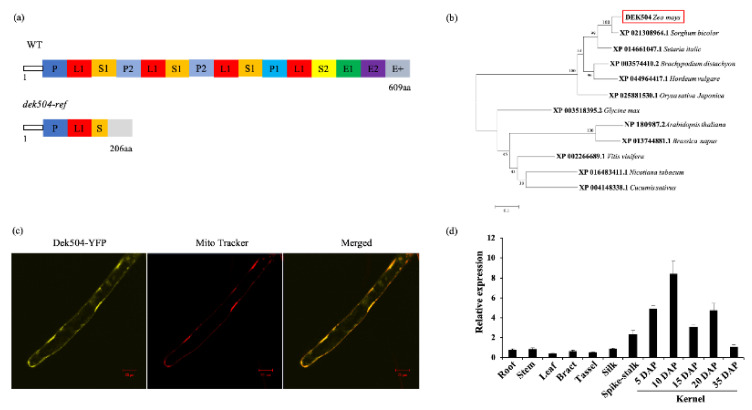
Maize *Dek504* encodes a mitochondrion-targeted E+-type pentatricopeptide repeat (PPR) protein. (**a**) Schematic diagram of the maize DEK504 protein harboring 15 PPR domains (P, L, and S domains) and C-terminal E1, E2, and E+ motifs as well as of *dek504-ref* with only one PPR domain. (**b**) Phylogenetic relationships of DEK504 and its homologs in *Cucumis sativus*, *Nicotiana tabacum*, *Vitis vinifera*, *Glycine max*, *Brassica napus*, *Arabidopsis thaliana*, *Oryza sativa Japonica*, *Hordeum vulgare*, *Brachypodium distachyon*, *Setaria italica*, and *Sorghum bicolor*. Numbers at the nodes represent the percentage of 1000 bootstraps. (**c**) Subcellular localization of the DEK504 protein in *Arabidopsis* roots. The yellow auto-fluorescent signal was generated by DEK504-YFP, and the red fluorescence of the mitochondria was detected using MitoTracker. Bar = 20 µm. (**d**) Expression profiles of *Dek504* in various organs and maize kernels at different developmental stages. *ZmGAPDH* was used as the reference gene. Values represent the means of three technical replicates, and the error bars represent SE.

**Figure 5 ijms-23-02513-f005:**
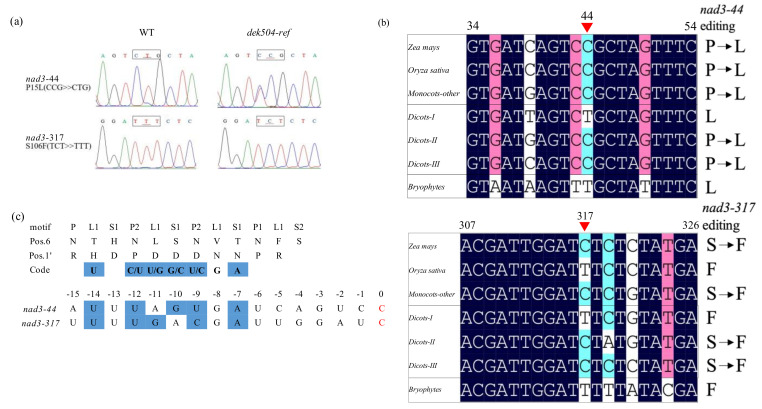
DEK504 is required for *nad3*-44 and *nad3*-317 C-to-U editing in maize mitochondria. (**a**) Analysis of RNA editing in *nad3* transcripts of wildtype (WT) and *dek504-ref* kernels. The editing sites are indicated by the red lines. Codons harboring the edited nucleotides and coded amino acids are shown. (**b**) Alignment of amino acid residues at positions 6 and 1′ in each PPR motif of DEK504 with a −1 to −15 bp upstream sequence of these four defective editing sites. Nucleotides matching the recognition code of DEK504 in nad3 are shown in blue. The edited C residues in nad3 are shown in red. (**c**) Alignment of the neighboring gDNA sequences of nad3. The gDNA and cDNA sequences were obtained from the NCBI database. Monocots included *Zea luxurians*, *Hordeum vulgare*, *Sorghum bicolor*, *Triticum aestivum*, *Bambusa oldhamii*, *Avena sativa*, *Phoenix dactylifera*, *Stipa capillata*, *Allium cepa*, and *Coix lacryma-jobi*. Dicots-I included *Arabidopsis thaliana*, *Brassica juncea*, *Brassica rapa*, *Brassica napus*, and *Boechera stricta*. Dicots-II included *Gossypium trilobum*, *Lagerstroemia indica*, *Bombax ceiba*, *Fagus sylvatica*, *Manihot esculenta*, *Heuchera parviflora*, *Malania oleifera*, and *Damnacanthus indicus*. Dicots-III included *Cannabis sativa*, *Panax vietnamensis*, *Panax notoginseng*, and *Tamarindus indica.* Red triangles indicate the positions affected by *nad3*-44 and *nad3*-317.

**Figure 6 ijms-23-02513-f006:**
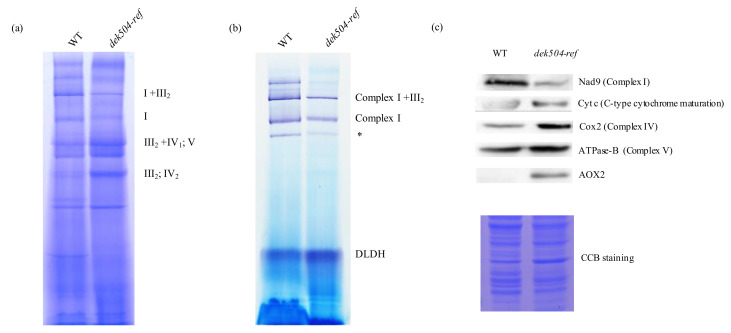
*dek504* mutation affected the assembly and activity of mitochondrial complex I. (**a**) BN-PAGE of the mitochondrial complexes. The positions of super complex I+III_2_, complex I, complex III, and complex IV are indicated. (**b**) In-gel NADH dehydrogenase activity assay of complex I. The positions of super complex I+III_2_ and complex I are indicated. DLDH was used as the loading control with the asterisk (*) representing the partially assembled complex I. (**c**) Western blotting with antibodies against NAD9, Cytc, Cox2, ATPase-B, and AOX.

**Figure 7 ijms-23-02513-f007:**
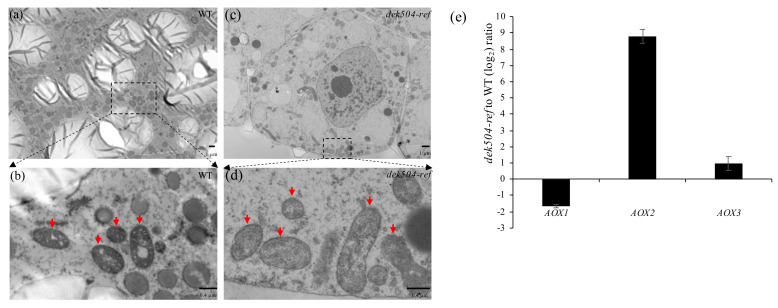
Mitochondrial morphology and *AOX* expression in the *dek504-ref* mutant. (**a**−**d**) Ultrastructure of developing endosperm in wildtype (WT) (**a**, **b**) and *dek504-ref* (**c**,**d**) kernels at 14 days after pollination (DAP). (**a**,**c**) Bar = 1 µm. (**b**,**d**) Bar = 0.4 µm. Red arrows indicate the mitochondria. (**e**) Quantitative real-time polymerase chain reaction (qRT-PCR) analysis of *AOX1*, *AOX2*, and *AOX3* expression in WT and *dek504-ref* mutant kernels. *ZmActin* was used as the internal control. Values are presented as means ± SE of three biological replicates. *AOX*, alternative oxidase.

## Data Availability

All of the data generated or analyzed during this study are included in the published article.

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
