# Peer review of "Dek504 Encodes a Mitochondrion-Targeted E+-Type Pentatricopeptide Repeat Protein Essential for RNA Editing and Seed Development in Maize"

_ijms, 2022, doi:10.3390/ijms23052513_

Round 1
Reviewer 1 Report
Dear Authors!
I enjoyed reading your manuscript. The manuscript is scientificaly sound, the design of the experiments and presentation are of high quality. I appreciate attacheing original gel imanges and all the supplementary data that nicely complement the data presented in the main file. The findings of your study are very interesting to me, working with maize. I have several comments - have found some grammatic errors in the file, please make sure that there are no grammatic errors in the file. All in all I find the study interesting and nicely and clearly presented.

Author Response
Dear reviewer,
Many thanks for your kindly consideration for publishing our work in International Journal of Molecular Sciences, as well as the constructive suggestions for improving the quality of this manuscript. We have revised this manuscript according to these suggestions. Please see the details in the flowing contents and also the corresponding changes marked with red font in the revised manuscript.
Responses to Reviewer comments:
Minor points:
- - Lane 39: "In 2005, CRR4, which possesses...". The paper that this statement refers to it was published in 2006, therefore change to: ""In 2006, CRR4, which possesses..."
Answer (A): Thanks. We have reviewed the previous reports about CRR4. CRR4 was first reported in 2005 by Emi Kotera et.al in “A pentatricopeptide repeat protein is essential for RNA editing in chloroplasts” published in Nature. We have corrected the reference.
- - Lane 40: "..... a pentapeptide repeat (PPR) motif, was found to be essential ..." Change to: "..... a pentatricopeptide repeat (PPR) motif, was found to be essential ..."
A: Thanks. We have corrected the spelling mistake from “pentapeptide” to “pentatricopeptide”.
- - Lane 49: the sentence: "....the canonical P motif a long (L, 35–36 aa) motif......." needs a coma: ".....the canonical P motif, a long (L, 35–36 aa) motif......."
A: Thanks. We have changed “....the canonical P motif a long (L, 35–36 aa) motif.......” to “.....the canonical P motif, a long (L, 35–36 aa) motif.......”.
- - Lane 508, References: This whole section clearly needs to be more carefully prepared. I could not find many references cited in the text:
Covello and Gray, 1989
Hiesel et al., 1989
Okuda et al. 2006
Malbert et al., 2020;
A: Thanks. These four articles were lost because they were not properly formatted when quoted, we have corrected the mistake and updated the references.
Major points:
- - Lanes 100-101: "...and the embryo lacked the scutellum, shoot apex, and primary root primordium following maturation (Figure 1c). "
This sentence should be rephrased. Hand-made sections of whole kernels shown in Fig. 1c do not allow the reader to appreciate the described anatomical changes observed in embryos of the dek504 mutant relative to wt. Authors should refer in addition to Fig. 2b-e, since thin sections of the embryos portray more convincing images. In Fig. 2b-e, we can appreciate that, although delayed in development, dek504 mutants develop coleoptile, primary leaves, and primary root primordia, contrary to what was described by the authors in this sentence.
A: Thank you for your suggestion. Fig. 2b-e are more appropriate to describe the morphology of embryonic development. We have deleted the sentence “and the embryo lacked the scutellum, shoot apex, and primary root primordium following maturation (Figure 1c).”, and modified the following description from “Paraffin-embedded sections showed that WT embryos developed visible scutellum, coleoptile, leaf primordium (LP), shoot apical meristem (SAM), and root apical meristem (RAM) at 12 DAP (Figure 2b), whereas dek504 embryos were arrested and remained at the transition stage, without visible differentiation in LP, SAM, and RAM. At 15 DAP, dek504 embryos reached the single LP stage (Figure 2c), but SAM and RAM did not appear until 18 DAP (Figure 2d, e). In addition, compared with WT kernels, which had starch-filled endosperms, dek504 kernels had small and underdeveloped endosperms, with a gap observed between the endosperm and seed coat at 12 DAP (Figure 2b), and the embryonic growth was retarded after 15 DAP (Figure 2c, d, e).” to “Paraffin-embedded sections showed that WT embryos developed visible scutellum, coleoptile, leaf primordium (LP), and shoot apical meristem (SAM) reached the L2 (second leaf primordia) stage at 12 DAP (Figure 2b), whereas dek504 embryos were arrested and remained at the transition stage, without visible differentiation in LP, and SAM. The WT embryo developed 3-4 leaf primordial, and complete structures with clearly visible SAM and root apical meristem (RAM) at 15 and 18 DAP, reaching to late embryogenesis stage (Figure 2c, d, h). By contrast, dek2849 embryos reached the L1 stage with one leaf primordium and SAM, but without RAM at 15 DAP. At 18 DAP, the mutant embryos remained at L1 stage and showed no significant sign of RAM development (Figure 2c, d, e). In addition, compared with WT kernels, which had starch-filled endosperms, dek504 kernels had small and underdeveloped endosperms, with a gap observed between the endosperm and seed coat at 12 DAP (Figure 2b). The dek504 endosperm development was slower than that of the WT at 15 and 18 DAP (Figure 2c, d).”
- - Lanes 230-232: " ......editing at nad3-44 (from 80% to 40%) and nad3-317 (from 94% to <1%) was dramatically decreased in dek504 kernels. "
To really confirm whether or not dek504 mutants decrease editing of nad3 transcripts at positions 44 and 317, authors should compare the mitochondrial DNA sequences of those genes in the dek504 mutant and wt maize to rule out possible polymorphisms (SNPs) in their mitochondrial genomes at the nad3-44 and nad3-317 sites. This is important because the dek504 mutant was derived from line 13d3147 and wt alleles most likely come from B73.
A: Thank you for your suggestion. To exclude possible polymorphisms (SNPs) at the nad3-44 and nad3-317 sites in the dek504 mutant and WT mitochondrial genomes, DNA of 10 kernels were extracted from dek504, WT-ref, and B73 each. A 357 bp DNA fragment of the nad3 region was amplified using specific primers (nad3-F and nad3-R). DNA fragments were sequenced separately for each sample by ABI 3730XL sequencer. The results showed that the nucleotides were the same at the nad3-44 and nad3-317 sites among dek504 mutant, WT-ref and B73 mitochondrial genome.
In addition, the mitochondrial genome is inherited from maternal parent in maize. The editing efficiency at nad3-44 and nad3-317 sites were analyzed between dek504 mutant and WT sibling kernels with the same mitochondrial genomes. Theoretically, the nucleotides were the same at nad3-44 and nad3-317 sites between the dek504 mutant and WT-ref. Therefore, the statistics of editing efficiency in the manuscript are reasonable.
Sincerely,
Yu Cui,
Feb 10, 2022

Reviewer 2 Report
In this manuscript Wang and colleagues made a very comprehensive characterization of a kernel defective mutant in maize, dek504. Through developmental, physiological, biochemical, genetical, and molecular analyses they concluded that dek34 encodes a mutant form of an E+-type PPR (pentatricopeptide repeat) protein involved in mitochondrial transcript editing, specifically in the edition of nad3 transcripts at positions 44 and 317, causing C to U changes. They later showed that these transcript modifications prevent the assembly of the mitochondrial complex I, explaining the defects and delayed development of dek504 mutant kernels. This is a very well written manuscript for the most part; it is clear and concise. The findings are novel, because they found a new PPR encoding gene (DEK534) that post-transcriptionally modifies nad3 transcripts at new positions resulting in amino acid changes that potentially cause functional defects to the NADH dehydrogenase 3 protein. I believe that this manuscript is worthy of publication in the International Journal of Molecular Sciences, but I also consider that it requires some improvement before it is accepted for publication.
Minor points:
- Lane 39: "In 2005, CRR4, which possesses...". The paper that this statement refers to it was published in 2006, therefore change to: ""In 2006, CRR4, which possesses..."
- Lane 40: "..... a pentapeptide repeat (PPR) motif, was found to be essential ..." Change to: "..... a pentatricopeptide repeat (PPR) motif, was found to be essential ..."
- Lane 49: the sentence: "....the canonical P motif a long (L, 35–36 aa) motif......." needs a coma: ".....the canonical P motif, a long (L, 35–36 aa) motif......."
- Lane 508, References: This whole section clearly needs to be more carefully prepared. I could not find many references cited in the text:
Covello and Gray, 1989
Hiesel et al., 1989
Okuda et al. 2006
Malbert et al., 2020;
Major points:
- Lanes 100-101: "...and the embryo lacked the scutellum, shoot apex, and primary root primordium following maturation (Figure 1c). "
This sentence should be rephrased. Hand-made sections of whole kernels shown in Fig. 1c do not allow the reader to appreciate the described anatomical changes observed in embryos of the dek504 mutant relative to wt. Authors should refer in addition to Fig. 2b-e, since thin sections of the embryos portray more convincing images. In Fig. 2b-e, we can appreciate that, although delayed in development, dek504 mutants develop coleoptile, primary leaves, and primary root primordia, contrary to what was described by the authors in this sentence.
- Lanes 230-232: " ....editing at nad3-44 (from 80% 231 to 40%) and nad3-317 (from 94% to <1%) was dramatically decreased in dek504 kernels. "
To really confirm whether or not dek504 mutants decrease editing of nad3 transcripts at positions 44 and 317, authors should compare the mitochondrial DNA sequences of those genes in the dek504 mutant and wt maize to rule out possible polymorphisms (SNPs) in their mitochondrial genomes at the nad3-44 and nad3-317 sites. This is important because the dek504 mutant was derived from line 13d3147 and wt alleles most likely come from B73.
Author Response
Dear reviewer,
Many thanks for your kindly consideration for publishing our work in International Journal of Molecular Sciences, as well as the constructive suggestions for improving the quality of this manuscript. We are really sorry for these language mistakes and greatly thank you to point out them. We have revised this manuscript according to these suggestions. Please see the details in the flowing contents and also the corresponding changes marked with red font in the revised manuscript.
Responses to Reviewer comments:
- Line 13
Answer (A): Thanks. We have changed “deaminate” to “deaminates”.
- Line 16
A: Thanks. We have corrected from “embryogenesis and endosperm development which produces small and collapsed kernels” to “embryogenesis and endosperm development which produce small and collapsed kernels”.
- Line 19
A: Thanks. We have changed “at the nad3-317 and nad3-44 site” to “at the nad3-317 and nad3-44 sites ”.
- Line 22
A: Thanks. We have changed “the amino acids encoded by editing nad3-44 and nad3-317 are highly conserved in monocots and eudicots” to “the amino acids are highly conserved in monocots and eudicots”.
- Line 23
A: Thanks. As your opinion, we have changed “whereas the events of C-to-U editing is not conserved in flowering plants.” to “whereas the events of C-to-U editing are not conserved in flowering plants.”.
- Line 26
A: Thanks. We have changed “nad3” to “NAD3”.
- Line 32
A: Thanks. We have changed “that predicted based on the genome sequence” to “that encoded from the genome sequence”.
- Line 33
A: Thanks. We have change “the editing-encoded amino acids” to “the amino acids encoded by edited mRNAs”.
- Line 39-1
A: Thanks. We have corrected from “editing target” to “editing targets”.
- Line 39-2
A: Thanks. We have added the full name of CRR4 as “Chlororespiratory Reduction 4 (CRR4)”.
- Line 49
A: Thanks. We have added a coma and changed “......the canonical P motif a long (L, 35–36 aa) motif.......” to “......the canonical P motif, a long (L, 35–36 aa) motif.......”.
- Line 52
A: Thanks. “whereas PLS-PPR proteins are almost exclusively related to C-to-U conversion” . This sentence may use the correct grammar. I will confirm with the scientific editor whether it needs to be modified.
- Line 69
A: Thanks. We have changed “loss function” to “loss-of-function”.
- Line 74
A: Thanks. “electron transport chain (ETC)” was first used in line 63. The abbreviation may be appropriate in line 74.
- Line 79
A: Thanks. We have changed “PPR proteins play pivotal roles in mitochondrial RNA editing to maintain its function during embryonic and endosperm development.” to “PPR proteins play pivotal roles in RNA editing to maintain mitochondrial function during embryonic and endosperm development.”.
- Line 96
A: Thanks. We have changed “Contrary to the wildtype (WT)” to “Contrary to the wildtype (WT) maize”.
- Line 113
A: Thanks. We have changed “the dek504 mutation delays plant growth and seed development and lowers seed nutrient reservoir accumulation.” to “the dek504 mutation delayed plant growth and seed development and lowered seed nutrient reservoir accumulation.”.
- Line 138
A: Thanks. We have corrected from “shown” to “showed”.
- Line 145
A: Thanks. We have corrected from “Magnified image of is shown (d)” to “Magnified image of fig.2d ”.
- Line 396
A: Thanks. We have changed “The maize plants were grown in an experimental field in Beijing or Hainan Province under natural conditions.” to “The maize plants were grown at Shunyi, experimental station of Institute of Crop Sciences, Beijing, China (116.60°E, 40.22°N) and Nanbin, experimental station of Institute of Crop Sciences, Sanya, Hainan Province, China (109.18°E, 18.36°N) under natural conditions.”.
- Line 442
A: Thanks. We have changed “GAPDH was used as the internal control.” to “Glyceraldehyde-3-phosphate dehydrogenase (GAPDH) was used as the internal control gene”.
- Line 447
A: Thanks. We have corrected from “Total RNA from each sample was extracted according using a commercial kit (Tiangen, Beijing, China) following the manufacturer’s protocol.” to “Total RNA from each sample was extracted using RNAprep Pure Plant Kit (DP432, Tiangen, Beijing, China) following the manufacturer’s protocol”.
- Line 462
A: Thanks. We have changed “the PCR products were cloned into the pEASY ®-Blunt Simple Cloning Vector (TransGen, Beijing, China).” to “the PCR products were cloned into the pEASY ®-Blunt Simple Cloning Vector (TransGen, Beijing, China).”.
- Line 472
A: Thanks. We have corrected from “were used for western blotting” to “were used for Western blotting”.
- Line 487
A: Thanks. We have removed the Chinese characters from Figure S5.
- Line 491
A: Thanks. We have changed Table S1 caption in the excel file from “List of PPR proteins involved in RNA editing in maize” to “List of PPR proteins involved in RNA editing of maize organelles”.
Sincerely,
Yu Cui,
Feb 10, 2022
Round 2
Reviewer 2 Report
I reviewed the revised version of the ms and I am satisfied, for the most part, with the answers to my points of concern.
- Some changes made by the authors in response to reviewer # 1 and # 2 comments require orthographical and style corrections. These are just a few examples:
lane 41: "...taegets...." it should be changed to "....targets....."
lane 138: "....primordial..." change to: "....primordia....".
lane 157: "Magnified image of fig.2d...", change to "Magnified image of Figure 2d..."
In addition, although I am satisfied about the response of the authors, in their cover letter, to my second comment (point 5), I consider that they should include the contents of their response somewhere in the text of the manuscript, as it is a very important control experiment that strengthens the idea that the dek504 mutant is affected in a mitochondrial RNA editing activity. I suggest section 2.4 (Results) and 4.8 (Materials and Methods).
Author Response
Dear reviewer,
We are really sorry for these language mistakes and greatly thank you to point out them. We have revised this manuscript according to these suggestions. Please see the details in the flowing contents and also the corresponding changes marked with red font in the revised manuscript.
Responses to Reviewer comments:
- lane 41: "...taegets...." it should be changed to "....targets....."
Answer (A): We have corrected ” ...taegets....” to "....targets.....".
- lane 138: "....primordial..." change to: "....primordia....".
A: We have corrected "....primordial..." to "....primordia....".
- lane 157: "Magnified image of fig.2d...", change to "Magnified image of Figure 2d..."
A: We have changed "Magnified image of fig.2d..." to "Magnified image of Figure 2d...".
- In addition, although I am satisfied about the response of the authors, in their cover letter, to my second comment (point 5), I consider that they should include the contents of their response somewhere in the text of the manuscript, as it is a very important control experiment that strengthens the idea that the dek504mutant is affected in a’ mitochondrial RNA editing activity. I suggest section 2.4 (Results) and 4.8 (Materials and Methods).
A: Thank you for your suggestion. We have added a sentence “The DNA nucleotides of nad3-44 and nad3-317 sites were the same between dek504 and WT kernels, excluding the possibility that polymorphisms (SNPs) affect editing efficiency.” in line 245 of this revised manuscript.
We have added the sentence “To exclude possible polymorphisms (SNPs) at the nad3-44 and nad3-317 sites in the dek504 mutant and WT mitochondrial genomes, DNA of 10 kernels were extracted from dek504, WT-ref, and B73 each. A 357 bp DNA fragment of the nad3 region was amplified using specific primers (nad3-F and nad3-R). DNA fragments were sequenced separately for each sample by ABI 3730XL sequencer.” in line 482 of this revised manuscript.
Sincerely,
Yu Cui,
Feb 17, 2022